# A Model for the Formation of Beliefs and Social Norms Based on the Satisfaction Problem (SAT)

**DOI:** 10.3390/e27040358

**Published:** 2025-03-28

**Authors:** Bastien Chopard, Franck Raynaud, Julien Stalhandske

**Affiliations:** Computer Science Department, University of Geneva, 7 Route de Drize, 1227 Carouge, Switzerland; franck.raynaud@unige.ch (F.R.); julienstalhandske@gmail.com (J.S.)

**Keywords:** sociophysics, belief systems and norm formation, satisfaction problems (SAT), cognitive model

## Abstract

We propose a numerical representation of beliefs in social systems based on the so-called SAT problem in computer science. The main idea is that a belief system is a set of true/false values associated with claims or propositions. Each individual assigns these values according to its cognitive system in order to minimize logical contradictions, thus trying to solve a satisfaction problem. Social interactions between agents that disagree on a proposition can be introduced in order to see how, in the long term, social norms and competing belief systems build up in a population. Among other metrics, entropy is used to characterize the diversity of belief systems.

## 1. Introduction

One goal of sociophysics is to propose methods inspired by statistical physics, mathematics or computer science to study social systems. The description of human features such as beliefs and opinions, in terms of formal concepts, leaves room for different approaches.

While sociophysics has emerged as a new discipline already several years ago [1], the metaphor of spins has been widely used to represent the opinion of individuals as a discrete Boolean variable [2], or to describe a strategic position of a person or of a political body [3,4]. Interactions between such entities can be implemented in analogy with the famous Ising spin model [2], or through many variants of the voter model [5,6,7] in which individuals switch their binary opinion according to that of their neighbors.

Although such an approach, which provides a high level of abstraction, may favor the discovery of very robust and general behaviors [8,9], it may appear disconnected from our own perception of what characterizes a human being whose behavior is driven by various elements, sometimes in synergy, sometimes in contradiction.

An interesting way to bring a description of human belief systems with more intuitive concepts was proposed in [10]. There, the authors consider a model of belief evolution under cognitive coherence and social conformity. In their model, an individual is represented as a network of *beliefs* and *concepts*. Concepts typically refer to an object, a person, an activity and so on. A belief is then a *positive* or *negative* association between two concepts. Often, three concepts *A*, *B* and *C* are found to be in such pairwise associations. For instance, if (A,B) and (B,C) are in a positive association, it is expected that so does (A,C). But, if it is not the case, there is frustration in the cognitive system of the individual which is interpreted as an increase in energy and a motivation to modify the cognitive network.

Their model also includes a social interaction based on the alignment of beliefs between two agents in a fixed network topology. The dynamics of the belief systems of the population is then governed by a balance between the energies due to internal coherence and social conformity. Among other results, this study computes a phase diagram where the size of the largest belief-homogeneous subpopulation is given as a function of the main model parameters (internal coherence, social interaction and “temperature”).

In the present study, we propose a different picture of a belief system, which is also characterized by its cognitive coherence and can be subject to social interactions. We consider that a belief system is a set of values TRUE or FALSE that each individual assigns to the same given set of propositions. The values given to these propositions must satisfy some cognitive logical constraints that represent links (or arguments) between the propositions.

For instance, consider the following three beliefs:P1: Climate change is detrimental to our environment;P2: Climate change is due to CO_2_ emission;P3: It is necessary to reduce fossil fuel.

A given individual may have assigned different values to these propositions according to some personal reasons. But there are logical implications between these propositions that any individual is assumed to understand. This is, for instanceIF(P1 AND P2)THEN P3
According to Boolean algebra, such an implication is TRUE unless the AND clause is TRUE and the conclusion is FALSE.

Thus, an individual who believes that P1 and P2 are TRUE should also believe that P3 is TRUE. Otherwise, its belief system is inconsistent. On the other hand, another individual who thinks that P2 is FALSE is consistent even if its belief is that P3 is FALSE. In this example there are three propositions with two possible values each, that is $2^{3}=8$ possible belief systems. They are all consistent except the system TRUE-TRUE-FALSE.

In short, a belief system is the choice of values associated with a given set of *N* propositions, constraint by a set of *M* Boolean equations among these propositions. In computer science, this structure defines what is known as a *satisfaction problem*, often abbreviated as SAT problem. The standard formulation of a SAT problem is to determine whether the *M* Boolean equations admit a solution. A similar formulation is to find an assignment of the *N* propositions which *minimizes* the number of unsatisfied equations. It is therefore natural to define this minimal value as the *energy E* of the belief system. A fully coherent system has energy E=0 as all the equations are satisfied. For totally inconsistent beliefs, E=M, meaning that all Boolean equations are unsatisfied with the chosen values of the *N* propositions.

Monasson et al. [11,12] have studied the existence of solutions to a family of SAT problems as a function of α=M/N. They showed that the probability that a random SAT problem is satisfiable exhibits a first-order phase transition for a critical value $\alpha_{c}$, when *N* and *M* go to infinity. Other quantities related to this problem are also subject to phase transition (see, for instance, [13]).

In the model we propose here, each individual (or agent) is characterized by its beliefs, namely the values chosen for the *N* propositions. These propositions are connected by a set of *M* logical relations, defining a cognitive system that can be shared by all agents or assigned individually to each of them. In the present work, we explore different types of scenarios, the implementation of which will be detailed later.

In our model, we can also consider a social interaction between pairs of agents. Such an interaction can be defined in several ways and have several possible outcomes. The simplest idea is to let agents whose belief systems are close enough discuss the value of a proposition for which they disagree. By agents that are close enough, we mean agents whose Hamming distance between their belief systems is less than a given threshold. The Hamming distance is the number of beliefs (or propositions) for which two agents have chosen a different value. In social science, this capability to discuss with people having similar opinions is called cognitive distance [14,15,16].

For instance, if the two agents disagree on the value of proposition $P_{i}$, the interaction will tend to make them accept a common value. Several mechanisms can be considered to determine which agent will impose its opinion on the other, based on some confidence level, or trust that is associated with the thought of the agent. The agent with the higher confidence level may initiate and possibly win the debate.

Among the ways to define a confidence level, we consider the following process. Let us consider that proposition $P_{i}$ appears in *m* out of the *M* equations. These equations can be called *arguments* as they influence the value of one proposition as a function of the others ones. Among the *m* arguments in which proposition $P_{i}$ is present, let us assume that $m_{+}$ of them are satisfied and $m_{-}=m-m_{+}$ are not. The values of $m_{+}$ and $m_{-}$ may differ between the two agents because of their different beliefs. Therefore, we can decide that the agent for which $m_{+}-m_{-}$ is the largest is the one with stronger arguments in favor of its value of proposition $P_{i}$. The opponent will accept to change the value of this belief provided the change is acceptable with respect to the variation in its internal coherence (i.e., its energy *E*). Typically, a Metropolis decision rule (see Equation (Equation 6)) can be used to accept or reject the new value. Note that in the voter models, or that of ref. [10] the agent that is considered for an opinion change is chosen at random, and not based on an asymmetry of coherence.

Note that in the above mechanism, only the value of a proposition is exchanged among the opponents. Section 3.3 will consider the case where arguments (i.e., cognitive relations) are also exchanged during the debate, the winner imposing its arguments on the opponent.

The formation of compatible or incompatible belief systems can then be studied as a function of the social interactions and the tolerance of individuals to internal contradictions. In Section 3, we consider the case of individuals, each with *N* beliefs initialized at random but with some internal coherence. We then study the possible coexistence of social norms in the population, that is, the occurrence of different mutually incompatible, stable belief systems.

Figure 1 gives an illustration of the difference between the representation of beliefs in three cases (spins as in the voter-like models, network of positive or negative associations between concepts as proposed in [10], and the present, SAT-based belief model).

In this paper, our first goal is to explore the above idea and show that it brings novelty and has potential for the modeling of social systems. We consider here a specific instance of our approach, leaving for future investigations more systematic studies, a deeper exploration of the richness of the model and applications related to real social data.

It turns out that social science proposes several interesting approaches to characterize the coherence, acceptation and rejection of hypotheses by human beings. It is beyond our knowledge and beyond the scope of this paper to present a comprehensive picture of this domain through the prism of social science but, clearly, models that would be known and accepted by both the sociophysics community and social scientists would bring an important synergy to the field. Here, we point out some references that support our approach and give a broader context. It is worth acknowledging the work on the so-called constraint satisfaction network (CNS) proposed by Thagard [17,18] and used by several authors considering experiments on the decision-making process and strategy of choices (see for instance [19,20]). In short, CSNs are simple networks of concepts that either excite or inhibit one another, a mechanism that reminds us of S. Kauffman genetic regulatory network [21]. CSNs can be viewed as a simpler version of ref. [10].

The logical contradictions that may appear in our SAT formulation can be related to the concept of cognitive dissonance in social science. A debate is whether human decisions precede or follow the establishment of coherent thinking (see for instance [22]). Experiments [23] have suggested that decision takes place as soon as the coherence of one possible explanation is discovered.

The concept of energy minimization, common in sociophysics, has also been considered in the physiology of the brain (see, for instance, [24]), and a large body of the literature related to this question exists. There, energy should not be viewed as a thermodynamic quantity but rather as an information-theoretic value, arising from population evolution and selection.

## 2. Detailed Model Description

In the previous section, we presented the general ideas of our model and the applications we would like to consider. Yet, in practice, several details should be specified. Here, we present a specific implementation of our model that we used to produce the results of Section 3.

### 2.1. The Belief and Cognitive Systems

As previously discussed, a belief system is the set of values assigned to *N* given propositions $P_{i}$. A cognitive system is defined as a set of *M* Boolean equations that link the *N* propositions through logical constraints and form a SAT problem.

Here, we consider a simplified type of cognitive system where only two propositions are involved in each equation. This is often called a 2-SAT problem. Such a system can be represented as a N×N matrix *G* where the element G[i][j] describes the Boolean relation between propositions *i* and *j*.

Here, we consider the following six Boolean operators(1)$$\begin{array}{ccc} L_{1}\left(P,Q\right)=\left(P\;\mathrm{XOR}\;Q\right) & \; & L_{2}\left(P,Q\right)=\left(P\;\mathrm{AND}\;Q\right)\\ L_{3}\left(P,Q\right)=\left(P\;\mathrm{OR}\;Q\right) & \; & L_{4}\left(P,Q\right)=\left(P\Rightarrow Q\right)\\ L_{5}\left(P,Q\right)=\mathrm{NOT}\;\left(P\Rightarrow Q\right) & \; & L_{6}\left(P,Q\right)=\left(P\Leftrightarrow Q\right) \end{array}$$
whose truth tables are(2)$$\begin{array}{ccc} \begin{array}{ccc} P & Q & L_{1}\\ T & T & F\\ T & F & T\\ F & T & T\\ F & F & F \end{array}\; & \begin{array}{ccc} P & Q & L_{2}\\ T & T & T\\ T & F & F\\ F & T & F\\ F & F & F \end{array} & \;\begin{array}{ccc} P & Q & L_{3}\\ T & T & T\\ T & F & T\\ F & T & T\\ F & F & F \end{array}\\ \begin{array}{ccc} P & Q & L_{4}\\ T & T & T\\ T & F & F\\ F & T & T\\ F & F & T \end{array}\; & \begin{array}{ccc} P & Q & L_{5}\\ T & T & F\\ T & F & T\\ F & T & F\\ F & F & F \end{array} & \;\begin{array}{ccc} P & Q & L_{6}\\ T & T & T\\ T & F & F\\ F & T & F\\ F & F & T \end{array} \end{array}$$

In total, there are 16 Boolean functions that take two arguments. Our choice is somewhat arbitrary but it is based on usual Boolean operators and the fact that, in total, it provides a correct balance of TRUE and FALSE.

With the introduction of functions $L_{s}$, s=1,2,…,6, a cognitive system *G* is a N×N matrix with G[i][j]∈{0,1,…,6}, where the value G[i][j]=0 corresponds to an absence of relation between $P_{i}$ and $P_{j}$, whereas G[i][j]=s≠0 means that $L_{s}\left(P_{i},P_{j}\right)$ is required to be TRUE. Usually, elements G[i][i] are set to zero to avoid unsatisfiable conditions on the value of each $P_{i}$. On the other hand, we do not impose a symmetry on *G*. So, G[i][j] and G[j][i] appear as two simultaneous constraints for $P_{i}$ and $P_{j}$. Section 2.2 will show how to make sure that such a SAT problem has solutions.

The number *M* of non-zero elements of *G* defines the density ρ of the cognitive system. Since G[i][i]=0 and G[i][j]≠G[j][i], we have(3)$$\rho =\frac{M}{N(N-1)}$$

A fully coherent belief system is a set of values for $P_{1}$, $P_{2}$, *…*, $P_{N}$ such that all G[i][j] constraints are satisfied (i.e., a solution of the SAT problem). Partially coherent belief systems are characterized by the number of unsatisfied G[i][j] relations. This number is defined as the energy *E* of the belief system. Fully coherent belief systems have E=0.

### 2.2. Construction of the Cognitive System

To setup a simulation, we create at random a N×N cognitive matrix *G* according to a chosen density ρ of non-zero elements. To make sure this cognitive system admits fully coherent belief systems, we build *G* by first defining two solutions of the SAT problem, called ${\mathrm{beliefs}}_{1}$ and ${\mathrm{beliefs}}_{2}$.

These solutions can be represented as two *N*-element Boolean vectors.$${\mathrm{beliefs}}_{1}=\left[v_{1}^{\left(1\right)},\ldots ,v_{N}^{\left(1\right)}\right]\;{\mathrm{beliefs}}_{2}=\left[v_{1}^{\left(2\right)},\ldots ,v_{N}^{\left(2\right)}\right]$$
where $v_{i}^{\left(\ell \right)}$ is the value assigned to proposition $P_{i}$ in belief system *ℓ*.

In what follows ${\mathrm{beliefs}}_{1}$ and ${\mathrm{beliefs}}_{2}$ will be chosen at random, but ${\mathrm{beliefs}}_{1}$ will have a probability 0.8 that $v_{i}^{\left(1\right)}=\mathrm{TRUE}$ and the opposite for ${\mathrm{beliefs}}_{2}$, for which $v_{i}^{\left(2\right)}=\mathrm{TRUE}$ with probability 0.2. The motivation for this approach is to ensure that the SAT problem has at least two solutions with a significant Hamming distance. We can then study whether one of these belief systems will be selected by the population dynamics, or if they can coexist. Note that, in general, and in particular when the density ρ of arguments in *G* is small, we may observe more than the two imposed zero-energy solutions.

We construct a cognitive system *G*, compatible with ${\mathrm{beliefs}}_{1}$ and ${\mathrm{beliefs}}_{2}$, such that G[i][j]=s and the following relation is satisfied(4)$$L_{s}\left(v_{i}^{\left(1\right)},v_{j}^{\left(1\right)}\right)=L_{s}\left(v_{i}^{\left(2\right)},v_{j}^{\left(2\right)}\right)=\mathrm{TRUE}$$
Depending on the values of the $v_{i}^{\left(\ell \right)}$, there might be several possible *s*, or none. In the former case, *s* is chosen at random among these possibilities. In the latter case, G[i][j]=0, meaning the absence of constraint between $P_{i}$ and $P_{j}$.

Note that in our approach, the SAT problem expressed by *G* is built out of the specification of ${\mathrm{beliefs}}_{1}$ and ${\mathrm{beliefs}}_{2}$. By this construction, the SAT problem always has solutions, which allows us to use a rather large ratio α=M/N compared to the values considered in [12]. In Section 3, we consider typically α=54/15=3.6.

### 2.3. Agents and Population

The next component of the model is a population of N individuals, each of them characterized by a belief system$${\mathrm{agent}}_{\ell }=\left[b_{1}^{\left(\ell \right)},\ldots ,b_{N}^{\left(\ell \right)}\right]$$
where $b_{k}^{\left(\ell \right)}\equiv {\mathrm{agent}}_{\ell }\left[k\right]$ is the Boolean value that agent *ℓ* assigns to proposition $P_{k}$.

Each agent *ℓ* has its own cognitive system $A_{\ell }$, which could be different among agents. Here, we consider a particular case where $A_{\ell }$ is based on a copy of *G* defined earlier, with possibly a probability 1−p of omission and a probability *q* of error. Omission means that, with probability 1−p, $A_{\ell }\left[i\right]\left[j\right]=0$; otherwise, $A_{\ell }\left[i\right]\left[j\right]=G\left[i\right]\left[j\right]$. In case of error, the value of $A_{\ell }\left[i\right]\left[j\right]$ is chosen at random in {0,1,…,6} with probability *q*; otherwise, $A_{\ell }\left[i\right]\left[j\right]=G\left[i\right]\left[j\right]$.

At the start of the simulation, the N agents’ belief systems are generated at random, with the same probability to have $b_{k}^{\left(\ell \right)}$ TRUE or FALSE. Then, we consider a phase to create a level of internal coherence with respect to $A_{\ell }$.

This is obtained by applying the so-called random walk SAT (RWSAT) algorithm [12,13,25], which is a simple metaheuristic to solve an SAT problem. At each step of the algorithm, a non-satisfied equation $L_{s}\left(b_{i},b_{j}\right)=\mathrm{FALSE}$ is chosen at random and the values of the two propositions are modified to make this relation satisfied. According to truth table Equation (3), for each of the six functions $L_{s}$, there are pairs of values $\left(b_{i},b_{j}\right)$ that make it TRUE. One of these pairs is then selected at random.

This algorithm has no guarantee to converge since the correction of an unsatisfied equation can invalidate another, previously verified. However, for the parameters we consider here, this algorithm is enough to make each individual sufficiently coherent. So, here we are not considering perfectly consistent individuals, and some logical flaw in their belief system is acceptable.

### 2.4. Interactions

After initialization, the belief system of each agent will be confronted with social interactions. In our approach, we propose that social interactions between two individuals depend on the Hamming distance *D* separating their belief systems. The hypothesis is that two agents with too different set of values for the *N* propositions may not be able to discuss and argue. Therefore, we introduce an interaction range $r_{int}$ which is the maximum Hamming distance acceptable for two individuals to interact. This limit is known as the *cognitive distance* in social science [14,15,16]. If this distance is too high, sharing opinions becomes impossible because individuals do not understand each other.

Interactions take place between pairs of individuals. We define a Monte-Carlo step as the process which gives a chance to each agent to interact with another one. In practice, a Monte-Carlo step consists of generating N pairs of agents at random, one by one. Each pair may fail to produce an interaction if the Hamming distance *D* between them is more than $r_{int}$, or if the two agents agree on all *N* beliefs.

When an interaction is possible, the agents choose at random one of the propositions $P_{i}$ on which they disagree. Let us then say that ${\mathrm{agent}}_{\ell }\left[i\right]=1-{\mathrm{agent}}_{k}\left[i\right]$ where here TRUE = 1 and FALSE = 0. For each agent, the internal coherence of the value assigned to $P_{i}$ is computed as *m*, the balance of TRUE and FALSE arguments involving $P_{i}$(5)$$m^{\left(\ell \right)}=\sum_{j}\left(2L_{s}\left(b_{i}^{\left(\ell \right)},b_{j}^{\left(\ell \right)}\right)-1\right)$$
where $s=A_{\ell }\left[i\right]\left[j\right]$ and, again, $L_{s}=1$ if TRUE and 0 if FALSE.

Let us suppose that ${\mathrm{agent}}_{\ell }$ has more arguments in favor of its choice of $b_{i}^{\left(\ell \right)}$ than ${\mathrm{agent}}_{k}$: this means that $m^{\left(\ell \right)}>m^{\left(k\right)}$. Then, ${\mathrm{agent}}_{k}$ has a pressure to change the value associated with $P_{i}$ and take that of its opponent, namely $b_{i}^{\left(k\right)}=b_{i}^{\left(\ell \right)}$. This change is accepted in a Metropolis probability $p_{metropolis}$ related to the variation in energy(6)$$\Delta E=E_{k}\left(\ldots ,b_{i}^{\left(\ell \right)},\ldots \right)-E_{k}\left(\ldots ,b_{i}^{\left(k\right)},\ldots \right)$$
due to this change $b_{i}^{\left(k\right)}=b_{i}^{\left(\ell \right)}$.

Formally, the energy *E* is computed as$$E=\sum_{i,j}\left(1-L_{s}\left(b_{i},b_{j}\right)\right)$$

The Metropolis rule is then to accept the change with a probability$$p_{metropolis}=\exp\left(-\Delta E/T\right)$$
where *T* is a “temperature” whose role is to allow some internal deterioration of coherence.

This interaction mechanism does not assume the existence of a predefined social network. Interaction is based on the state of the agents (their belief system) not their position in a known acquaintance graph. As beliefs may change over time, the interaction network is dynamic.

Note that random numbers are used at several levels in this model. First in the construction of the cognitive matrix *G*, then in the initialization of population agents (beliefs and, depending on the scenario, part of the cognitive matrix $A_{i}$). Finally, there is an intrinsic stochasticity in the Monte Carlo steps which selects different sequences of agent-agent interactions. The first two elements correspond to the initialization of the system and can be reused in repeated simulations. The Monte-Carlo part could still lead to different outcomes. In Section 3, we will briefly discuss the reproducibility of the results.

### 2.5. Clusters

In order to analyze the evolution of the belief systems over time (i.e., over the number of Monte-Carlo steps), the N agents will be grouped in clusters according to the proximity of their beliefs.

Two agents will be in the same cluster if their Hamming distance is less or equal to a given threshold value $r_{cluster}$. In what follows, for the sake of simplicity, we chose $r_{cluster}=0$. Thus, a given cluster contains all agents having the exact same belief system.

A benefit of this choice is that each cluster can be uniquely identified by the decimal value obtained by interpreting the *N* Boolean values of the belief system as a *N*-bit number. Each cluster is then specified by its ID, a number between 0 and $2^{N}-1$.

This ID should not be confused with the cluster index which results in the way our algorithms determine the cluster. For instance, agent 0 is, by construction, always in the first-built cluster. Indeed, the algorithm works as follows: agents are visited sequentially. If an agent is not marked as already belonging to a cluster, it becomes the first member of the next cluster. Then, all other, yet unmarked, agents are visited and marked as neighbors of the considered cluster member. The neighbors of these neighbors are recursively marked. This process continues until all agents are associated with a cluster.

In Section 3, the results of the numerical simulations will be presented in terms of the emerging clusters, their size, energy and entropy.

## 3. Results: Coexistence of Belief Systems

In this section, we explore four scenarios through numerical simulations. We are interested in the time evolution of the belief systems of the population due to social interactions and internal coherence.

The four scenarios differ by the value of some model parameters. However, for all four scenarios, the following parameters are fixed.

Number of propositions: N=15;Population size: N=50;Distance range in cluster: $r_{cluster}=0$;Density of the cognitive system: ρ=0.3;Probability of error for the agent cognitive system: q=0.

In addition, we impose two fully rational, zero-energy belief systems(7)$$\begin{array}{ccc} {\mathrm{beliefs}}_{1} & = & [0,1,0,0,1,1,0,1,0,0,0,0,0,0,0]\;ID=178 \end{array}$$(8)$$\begin{array}{ccc} {\mathrm{beliefs}}_{2} & = & [1,1,1,1,1,1,1,1,0,0,1,1,1,1,0]\;ID=15615 \end{array}$$
The Hamming distance matrix $D({\mathrm{beliefs}}_{i},{\mathrm{beliefs}}_{j})$ between these two fully consistent, zero-energy solutions is:(9)$$D=\left(\begin{array}{cc} 0 & 8\\ 8 & 0 \end{array}\right)$$

The values of *N*, ρ, ${\mathrm{beliefs}}_{1}$, ${\mathrm{beliefs}}_{2}$ and the choice of a random number seed specify the cognitive matrix *G* shown in Equation (Equation 10). It contains 54 Boolean conditions (namely a fraction ρ=0.3 of the fully connected situation with $E_{max}=15\times 15=225$ relations, minus self relations and incompatible relations between the two imposed zero-energy solutions). In what follows, unless otherwise specified, the temperature is arbitrarily set to $T=0.1\times E_{max}=22.5$.

We are interested to determine how the 15 beliefs (propositions) of the 50 agents evolve over time, and whether several agents adopt the same belief systems.

Here, we consider the cognitive system *G* defined as(10)$$G=\left(\begin{array}{ccccccccccccccc} 0 & 0 & 4 & 4 & 0 & 3 & 0 & 0 & 0 & 0 & 0 & 0 & 6 & 0 & 0\\ 0 & 0 & 0 & 3 & 3 & 0 & 3 & 0 & 0 & 0 & 0 & 0 & 0 & 0 & 3\\ 0 & 0 & 0 & 0 & 0 & 3 & 4 & 0 & 0 & 0 & 6 & 0 & 4 & 0 & 0\\ 0 & 0 & 0 & 0 & 0 & 3 & 0 & 0 & 0 & 0 & 0 & 0 & 6 & 0 & 0\\ 3 & 0 & 0 & 0 & 0 & 0 & 3 & 0 & 5 & 0 & 0 & 0 & 0 & 3 & 5\\ 0 & 3 & 0 & 0 & 0 & 0 & 3 & 0 & 5 & 0 & 0 & 0 & 0 & 3 & 5\\ 0 & 0 & 0 & 0 & 0 & 0 & 0 & 4 & 0 & 0 & 0 & 0 & 0 & 0 & 0\\ 3 & 6 & 0 & 0 & 4 & 4 & 0 & 0 & 0 & 5 & 0 & 0 & 0 & 0 & 0\\ 0 & 0 & 0 & 0 & 0 & 0 & 0 & 4 & 0 & 0 & 0 & 0 & 0 & 4 & 4\\ 4 & 0 & 4 & 0 & 0 & 3 & 0 & 3 & 0 & 0 & 4 & 0 & 4 & 0 & 0\\ 6 & 0 & 0 & 0 & 0 & 4 & 6 & 0 & 0 & 0 & 0 & 0 & 0 & 6 & 0\\ 0 & 0 & 0 & 6 & 3 & 0 & 0 & 0 & 0 & 0 & 0 & 0 & 0 & 0 & 0\\ 4 & 0 & 0 & 0 & 4 & 0 & 0 & 0 & 0 & 0 & 0 & 0 & 0 & 0 & 0\\ 0 & 0 & 0 & 0 & 0 & 0 & 0 & 4 & 0 & 0 & 0 & 4 & 0 & 0 & 0\\ 0 & 4 & 0 & 0 & 1 & 1 & 0 & 1 & 0 & 0 & 0 & 0 & 0 & 4 & 0 \end{array}\right)$$

The numbers 1 to 6 refer to one of the six logical relations between propositions defined in Equation (Equation 1). An exhaustive search shows that the SAT problem defined by *G* has no other solutions than ${\mathrm{beliefs}}_{1}$ and ${\mathrm{beliefs}}_{2}$. The absence of $L_{2}$, the AND function, is not surprising since ${\mathrm{beliefs}}_{1}$ and ${\mathrm{beliefs}}_{2}$, which should both be solutions, have an opposit bias towards TRUE and FALSE.

The numerical experiment consists of generating a population of N=50 individuals, with initially fully random values for the 15 propositions. Then, each agent is subject to an improvement of its internal coherence, using 10 steps of the RWSAT algorithm described earlier. From this process, 10 agents have zero energy, corresponding to solution ID-15615 (6 agents) and solution ID-178 (4 agents). The energy of the other agents ranges between 1 and 18. The maximum possible energy is 54, obtained when no relation of the cognitive matrix is satisfied.

### 3.1. First Scenario: Fully Informed Agents and Full Interaction Range

In our first scenario, we study the behavior of the agents obtained with the additional constraints

Interaction range: $r_{int}=N=15$;Fraction of *G* copied to each agent: p=1 and thus $A_{\ell }=G$, ℓ=1,…,N.

The value p=1 means that the agents are fully informed. The value $r_{int}=15$ indicates that agents can interact with any other agent whose distance is less or equal to 15, that is all agents.

The simulation starts with 37 clusters defined by agents having the same beliefs (same value for the 15 propositions). Figure 2, left panel, shows the evolution of the number of clusters during 400 Monte-Carlo steps. The right panel shows the evolution of the entropy *S* of the belief systems, defined as(11)$$S\left(t\right)=\frac{1}{N}\sum_{i=1}^{N}p_{i}\ln p_{i}$$
where $p_{i}$ is the probability that proposition $P_{i}$ is true in the population, estimated as the number of times this proposition is true among the N agents(12)$$p_{i}=\frac{1}{N}\sum_{\ell }^{N}b_{i}^{\left(\ell \right)}$$

This entropy metric complements the data about the number of clusters and their sizes, as depicted in Figure 3. The figure also shows the Boolean values of the 15 propositions, the corresponding belief ID of the cluster in decimal form, and finally the energy of the cluster. The entropy gives additional information about the difference between the belief systems attached to these clusters, and the diversity of opinion among the agents. For instance, if the population contains many clusters, but with very similar belief systems, *S* will be small. On the other hand, a situation with two clusters that disagree on the value of most propositions will have a high entropy.

As shown in Figure 3, we observe that clusters 1, 2 and 4 form the largest partitions of the population. Cluster 3 has only two agents.

Whereas these clusters are mostly stable over time, we observe that, during time, agent 22 may change its mind about one proposition and occasionally form its own cluster. This is for instance the case at 380 Monte-Carlo steps. This movement between two clusters is illustrated in Figure 3 (upper-left panel) with a dashed arrow.

The Hamming distance between the above 5 clusters, at 380 Monte-Carlo steps, is given by the matrix(13)$$D=\left(\begin{array}{ccccc} 0 & 8 & 1 & 7 & 6\\ 8 & 0 & 9 & 1 & 2\\ 1 & 9 & 0 & 8 & 7\\ 7 & 1 & 8 & 0 & 1\\ 6 & 2 & 7 & 1 & 0 \end{array}\right)$$
We see that, except for the two zero-energy solutions ID-178 and ID-15615 that differ by eight beliefs, the other clusters are always close to one of these two. The entropy *S* of the population beliefs shown in Figure 2 (right panel, black curve) gives a similar indication about the diversity of the beliefs: a few clusters but a rather high entropy.

In the middle panel of Figure 3, we show the trajectory of two agents in the beliefs space, namely within all $2^{15}$ possible values of the 15 propositions. We see that agent 22 does not converge to a final cluster, but oscillates between two of them. On the other hand, agent 20 quickly reaches a zero-energy cluster. In this figure, we also show, with a black or a red circle, which other agent has been responsible for the change of opinion of agents 20 and 22, respectively.

Successful interactions quickly stop with agent 20, but keep going on with agent 22, thus explaining the fact that it does not reach a stable belief system. We can follow the trajectory of all agents in the lower panel of Figure 3.

Note that, when agents are fully informed (p=1), agents in zero-energy clusters cannot move out of their belief system because no agent from another zero-energy cluster has more arguments to induce the change in value of a controversial proposition. This is due to the fact that, for zero-energy solutions, $m_{+}-m_{-}$ are always the same when p=1.

The choice of the temperature *T* affects the number of successful interactions (those that caused a change of opinion). In this simulation, we observe that 400 successful interactions resulted in an increase in energy, while 616 interactions were decreasing it. Initially, the average energy of the population was 4.2 and it went down to 0.36 after 400 Monte-Carlo steps.

Table 1 shows the reproductiveness of the evolution of the belief systems in terms of the number of individuals in each emerging cluster. We consider here six runs with the same cognitive matrix *G*, but each time a new set of beliefs in the initial population.

We observe that clusters ID-178, ID-242, ID-15615 are selected in all six runs and are consistently populated across all repetitions. We also observe that a fourth cluster may be created, but not always the same (in two runs, cluster ID-16127, and ID-1266 in the other two). These additional clusters are less populated and have a varying size over time.

The persistence of the presence of energy-1 cluster ID-242 here and in the other scenarios can be understood as follows. It differs from ID-178 by the value of proposition $P_{6}$ but, due to the specific value of *G*, both clusters turn out to have the same value of $m_{+}-m_{-}$ for $P_{6}$. However, the value of $P_{6}$ for belief system ID-242 contradicts one condition of column 6 of *G*, hence its energy E=1.

The experiment can also be repeated with another cognitive system *G*, obtained from another sequence of random numbers. By construction, there are always two zero-energy solutions, but three or four fully coherent solutions are also frequent with ρ=0.3 and N=15. In the steady state, the population may or not contain all of them. For instance, if an agent is initialized with such a coherent belief system, it will survive as such an agent cannot change. On the other hand, the absence of a zero-energy agent does not prevent it from appearing out of the interactions. As an example, in a simulation where no internal coherence is imposed on the agents (no RWSAT steps), we have observed that all the agents (initially very inconsistent) evolve to only two zero-energy clusters out of the four existing ones.

The specific belief system of an individual has a clear impact on its capability to affect other agents’ beliefs during interaction. For instance, a single zero-energy individual will quickly destabilize an homogeneous, yet incoherent population, and restore consistency. A single sub-optimal agent also has an impact on a subpopulation of lesser coherence, by reducing the total energy of the population. Therefore our model confirms the fragility of a unique opinion that has conquered the entire population but whose coherence is no longer optimal.

Before considering our second scenario, we briefly discuss the effect of the value of N, the population size. Figure 2 (right) shows that, in terms of the diversity of belief systems, 50 or 500 individuals produce similar results, indicating that N=50 is already an adequate size to study the behavior of the model.

In terms of the size of the resulting clusters, Table 2 shows, for a single run, the size of the largest clusters. The observation is that, for N=500, more small clusters (1 to 3 individuals) appears than for N=50. However, all large clusters are the same in both cases, yet more populated when N increases. Up to expected fluctuation, a reasonable scaling of a factor of 10 is present between the two considered problem sizes.

### 3.2. Second Scenario: Limited Interaction Range

In the previous example, the interaction range was defined as $r_{int}=15$, meaning that agents that have up to 15 different opinions are ready to discuss and challenge their beliefs. With a total of 15 propositions, this means that anyone can interact with anyone. This will favor the exchange of ideas and may lead to a more homogeneous population. On the other hand, restricting the interaction range to individuals who only disagree on a few beliefs prevents agents with diverging beliefs from coming to an agreement. If individuals do not talk with people who are too different, there is less chance of reaching a common system of values.

The results of the simulation after 400 steps are illustrated in Figure 4 with *G* defined in Equation (Equation 10). The blue line in Figure 2 (left) shows the time evolution of the number of clusters, whereas the right panel of this figure shows the entropy evolution. It turns out that scenario 2 has the largest entropy of all cases considered here. This high entropy value reflects the existence of many clusters, each with rather different belief systems.

Compared to the previous case, more clusters are formed, 10 in the present run. Several of them contain only one or two individuals. The larger clusters are still the zero-energy ones but are less populated than before. Several clusters are stable with only a few individuals and a positive energy. In addition to generating more clusters of agents, this situation exhibits a faster convergence to a steady state, with clusters that are then totally stable in time.

This scenario is also characterized by much less successful interactions among pairs of agents. Driven by the Metropolis decision rule, one observes only 8 of them that increased agent energy against 100 that decreased it.

As for the previous scenario, the model is stochastic and different runs based on different sequences of random numbers will give different partitions of the population. However, the trend illustrated here, with the role of the interaction range $r_{int}$ on the number of clusters and their stability, is expected to be robust.

Table 3 shows the most robust belief systems obtained with the same cognitive matrix *G*, but different initial populations generated over six repetitions of the process. In the table, only the clusters that appear frequently are reported. In each of the six runs, about 10 stable clusters are generated, several with only one or two individuals, which change from one run to the next. We observe that clusters that were never selected when the interaction threshold was maximum are now present and rather robust as they keep appearing in most repetitions.

### 3.3. Third Scenario: Not Fully Informed Agents

In a third scenario, we consider agents that are not fully informed in the sense that their own cognitive matrix is a partial copy of the original one. Here, we take p=0.8, which means that each agent knows only about 80% of *G*. So, with probability 0.2 the element of G[k][ℓ] is not copied to $A_{i}$, agent *i* cognitive system. Instead, $A_{i}\left[k\right]\left[\ell \right]=0$. The indices *k* and *ℓ* for which this omission happens are chosen at random for each individual. Therefore, some relations between propositions are missing for some agents, which we interpret as the fact that these individuals may have fewer arguments than others to justify their opinion.

An interesting consequence of this partial information is that now an agent in a zero-energy cluster can convince agents from another zero-energy cluster. This is illustrated in Figure 2 with the red curve.

The other parameters of the simulation are identical to those that produced the black curve. Agents are then free to interact with anyone else ($r_{int}=15$). Since a change of opinion in our model is based on the number *m* of arguments in favor of the chosen value TRUE or FALSE of a proposition, agents in a zero-energy cluster that have only a few arguments to justify their choice may well be influenced by a “stronger” agent from another zero-energy belief system.

Figure 5 presents the two belief clusters formed during evolution. They are clusters of beliefs ID-178 and ID-242, which are almost identical as they differ only by one belief (Hamming distance 1). The energy of the agents in cluster ID-242 is E=1 for 21 of them, and E=0 for others 6. Note that since the cognitive matrix $A_{i}$ of each agent *i* may differ in the number of arguments between propositions, they may have different energy for the same set of beliefs.

The trajectories of the agents in the beliefs space are shown in the middle panel of Figure 5. The two clusters are totally stable and almost identical in terms of the beliefs that characterize them. Due to this small distance and the low entropy characterizing the population diversity (see Figure 2, red curve, right panel), we interpret this outcome by saying that the belief system ID-178 is absorbing.

In terms of the number of successful influences between agents subject to the chosen temperature, we observe 1674 interactions that increased energy, and 2473 that decreased it. The ratio between these two numbers is about 0.7, very similar to the ratio observed in scenario 1 where it is around 0.65. But it is much larger than what is observed in scenario 2, where this ratio is only 8/100. These numbers reveal again a difference between the three scenarios. A deeper investigation is yet needed to better understand the origin of these variations.

A conclusion suggested by this result is that several fully coherent belief systems can coexist in a stable way when the same information if shared by all agents. But one belief system may absorb the others if knowledge is partial and some agents have a more elaborate set of arguments to impose their beliefs.

Table 4 illustrates how the belief systems evolve when the process is repeated with the same cognitive matrix *G*, but an initial population generated by different sequences of random numbers, and thus possibly different *A*’s. A closer look at the dynamics of the agents in this scenario shows that some agents keep oscillating between clusters ID-178 and ID-242. This is the case of agent 0 whose trajectory in the beliefs space is shown in the lower panel of Figure 5. There, the blue dots represent the index of the agent that moved agent 0 from one cluster to the other. This agent is always a member of the other cluster, but not always the same, as clearly visible.

As already mentioned clusters ID-178 and ID-242 differ only by the value of proposition $P_{6}$, which is false in belief system 178 but true in belief system 242. Also, they are mutually robust to interactions when full knowledge is considered (p=1). The situation may change when p<1. It turns out that the cognitive matrix $A_{0}$ of agent 0 is such that proposition 6 does not influence the coherence of the other propositions, because row 6 of $A_{0}$ is accidentally null, due to the partial copy of *G*, with p=0.8. So, when interacting with another agent, for which proposition 6 has several positive arguments, agent 0 will easily change its mind. Such influencing agents may belong to any of the two clusters, and during the random interactions, chances are that such a change happens.

### 3.4. Fourth Scenario: Agents with Different Levels of Knowledge

Our model offers several possibilities to study various interaction scenarios between the agents. In this section, we investigate the impact of different knowledge levels in the population. By knowledge level we mean the number of arguments, or relations, that an agent has in its own cognitive matrix. As opposed to the previous examples, here each agent copies a different fraction of the original cognitive matrix *G*, where the fraction is chosen at random, uniformly, between 0 and *p* for each agent.

In the present scenario, we modify the interaction process used earlier in order to allow agents to share arguments. So, not only the convinced agent *i* adopts the value of a controversial proposition *k* from an opponent agent *j*, but it also adopts its arguments relative to the said proposition. Therefore, the number of arguments in agents’ cognitive matrices *A* may change, hence their level of knowledge.

The motivation for this numerical experiment is to observe how the knowledge level evolves over time. A question is for instance whether or not agents with fewer arguments will dominate the discussion in the long run. Is it true that, in real debates, people having simple arguments (not to say simplistic) but who look very clear and sure about their opinion will convince the silent majority, and overcome other actors having too many arguments to produce a simple message without nuance?

In a formal way, the interaction considered in this scenario amounts to copying the corresponding row and column of the cognitive matrix of the stronger agent *j*. In the formula, this is expressed as$${\mathrm{agent}}_{i}\left[k\right]={\mathrm{agent}}_{j}\left[k\right]\;A_{i}\left[.\right]\left[k\right]=A_{j}\left[.\right]\left[k\right]\;A_{i}\left[k\right]\left[.\right]=A_{j}\left[k\right]\left[.\right]$$
where $A_{\ell }$ is the cognitive matrix of agent *ℓ*.

In the case of such an interaction, we assume that both agents may possibly accept the above modifications, according to a Metropolis decision rule. In practice, each agent in the confrontation evaluates the modification of its energy ΔE that would be caused by adopting the beliefs and arguments of its opponent.

Figure 6 shows, for our example, the evolution of the number of clusters in this scenario, with two different ranges of interaction $r_{int}=15$ or $r_{int}=4$.

Starting with $r_{int}=4$, we again observe that such a limited interaction range produces stable sub-populations, yet usually more than one. Their composition is given in Figure 7. As each agent can have a rather different set of arguments and thus a different energy, we also list, for each individual, its energy and level of knowledge (the number of relations its cognitive matrix contains).

We observe one large cluster, with agents having energy between 0 and 2, and three smaller ones, also with an energy close to the optimum.

Initially, the agents have an average knowledge level of 25.86 with standard deviation of 16.18. In the stable regime, the average knowledge level decreases to 23.1, which is not a large modification, but the standard deviation dropped more significantly to 5.71. During the process, the average energy of the population changes from 1.54 down to 0.94.

On the other hand, with the maximum interaction range of $r_{int}=15$, only 1 cluster is obtained, containing all agents, with energy 0 or 1, and knowledge levels between 24 and 37.

In this simulation with $r_{int}=15$, the final average knowledge level increased from 25.86 to 28.48, with a change in standard deviation from 16.18 to 3.56. The energy changed from 1.54 down to 0.32, and its standard deviation decreased from 2.72 to 0.46. Figure 2 (right panel, green curves) shows the time evolution of the loss of diversity in the population belief systems. In the steady state regime, the entropy drops to zero. This is the case with both population sizes but the time scale is significantly larger for the population with N=500 individuals.

Beyond this partitioning of the initial population in different numbers of clusters, this example shows that the level of knowledge of the population has changed during the evolution. In both cases (interaction thresholds 4 or 15), the initial average of the population was 25.86, with some agents having more than 40 arguments supporting their 15 beliefs, down to agents owning only a few relations. All these agents were initially rather consistent with energy ranging between 0 and 12, with a population average of 1.54.

It turns out that other simulations with other random number sequences show that the amplitude of knowledge variation is probably not significant. Its increase or decrease is not related to the interaction range.

On the other hand, what seems clearly significant is the drop in the standard deviation of the values of the knowledge levels among the population. We observe a decrease of about a factor of 4, for both interaction ranges.

As already said, several agents initially have a high knowledge level (between 40 and 50 in this example). In the steady state, the maximum knowledge level is down to around 30. This can be interpreted as an impoverishment of knowledge in the population.

So, the proposed dynamics lead to homogenization and a simplification of the arguments in the population. This can be interpreted by saying that the debate evolves towards less complicated interdependence between propositions. Alternatively, this may indicate that simple arguments, without subtlety, invade the whole population.

A mechanism that explains this collective behavior at the level of the individuals is the fact that with less arguments in their cognitive matrix, the agents have more chance to be in a consistent state of low energy, just because there are less conditions to be satisfied.

## 4. Discussion and Conclusions

This paper explores a way to represent belief systems in a population of individuals. Belief systems are defined as a set of propositions that are evaluated as TRUE or FALSE by each individual. In addition to the set of propositions, logical arguments exist between these propositions so as to specify a cognitive coherence between them. This cognitive system is defined as a SAT problem, namely a set of Boolean equations that connect the propositions and that need to be satisfied by the chosen values of the propositions.

The coherence of a belief system is defined through the number of satisfied cognitive equations. In practice, incoherence is measured as the energy associated with the belief system. Energy is defined as the number of contradictions due to inconsistent choices of the value of the propositions with respect to the Boolean equations.

Social interactions can be defined among individuals with different beliefs, provided they can “understand” each other, that is if the distance between their belief systems is within a given Hamming distance threshold.

In this first study of this model, we focused on the dynamics of the belief systems within a population. We have considered four scenarios that differ by some parameter values (interaction range and partial or complete knowledge of the individuals) but yet with the same cognitive arguments in all four cases. Our main finding is that the population can be (1) divided among the most coherent belief systems; (2) a rather large number of belief systems coexist and are very stable, but not all very coherent; (3) a unique belief system can attract all individuals; (4) the level of knowledge in the population tends to homogenize.

The coexistence of several subpopulations with more or less incompatible belief systems is also present in [10]. However, the parameters used to describe the coexistence space are very different, so it is difficult to consider a quantitative comparison. Similarly, the possibility to destabilize a non-optimal homogeneous population is present in both approaches.

In our model, the confidence level associated with one belief could be defined in various ways. For example, to go beyond the balance of the number of supporting arguments studied here, a confidence level could be a number in [0,1] directly associated with each proposition, as an additional variable. Changing opinions will be easier if the confidence level is small. The value would decrease when interacting with an agent having a different opinion for the considered proposition, or increase if the two agents share the same point of view on the question.

In addition to the confidence level, a criteria of importance could be added to each individual’s belief so as to affect some propositions that, for cultural or personal reasons, cannot change their value.

Beyond the proposed scenarios, other interesting situations could be considered with our model. For instance, we could study the incremental construct of belief systems by adding new propositions one after the other. Individuals will accept or reject a new proposition in order to minimize the contradictions with previous beliefs, according to the logical arguments that connect the new proposition with the others. As a function of small consistent initial belief systems, the progressive formation of distinct social norms could be studied.

In a more prospective way, our model could possibly be applied to the problem of negotiation among opposite entities, in order to better design strategies of cooperation. Related to that, a very challenging problem would be to determine the beliefs of real persons or political groups out of their actions.

From the specific results presented in this study, several questions needing more investigations can be formulated. For instance, the topology of the agents’ network defined by the Hamming distance could be investigated for various thresholds.

Moreover, we can wonder what makes an inconsistent agent choose a particular zero-energy belief system over another? An SAT problem specifies an energy landscape on the space of proposition values, and we may conjecture that some solutions are more accessible in terms of energy barriers.

Such questions are related to the connection between disordered systems in statistical physics and optimization problems, links already mentioned in 1987 [26]. The minimization of contradictions in SAT problems is known to be NP-hard in computer science. It is not equivalent to a spin glass system, first because it may contain conditions with more than two variables. Even in the present case, where only two-proposition arguments are used, it is not reducible to a spin glass system because Boolean algebra is not equivalent to a multiplication.

On the other hand, the minimization of the number of contradictions in a set of *M* Boolean equations is close to the maximization of fitness in the so-called NK landscape problem proposed by S. Kauffman to describe a regulatory genetic networks [21]. NK problems are also commonly used as a benchmark in hard optimization problems [13]. The link with our problem is obtained by defining the fitness function as the sum of the values of the *M* equations (an equation which is TRUE has value 1 and a FALSE equation has value 0).

Solving SAT or NK problems exactly may require algorithms of exponential complexity that are not tractable when the number of propositions becomes large. Metaheuristics such as the RWSAT algorithm, or methods such as simulated annealing or genetic algorithms are known to give solutions of acceptable quality in an acceptable time. The interaction model we considered between individuals with different beliefs is close to those used in these nature-inspired processes and their efficiency is related to the energy landscape that shapes the evolution of belief systems in the space of propositions.

As an interesting illustration of the link between our problem and the way a metaheuristic explores a difficult energy landscape, we mention again that, when stating from random inconsistent agents (no RWSAT step), we observe just from social interactions an efficient convergence to the zero-energy solutions. This is similar to genetic algorithms that achieve good performance through a collective local optimization process. In our metaphor of a social system, it illustrates the differences that may exist between individuals improving their internal coherence in isolation and a population that reaches the same objective through a continuous exchange of ideas.

## Figures and Tables

**Figure 1 entropy-27-00358-f001:**
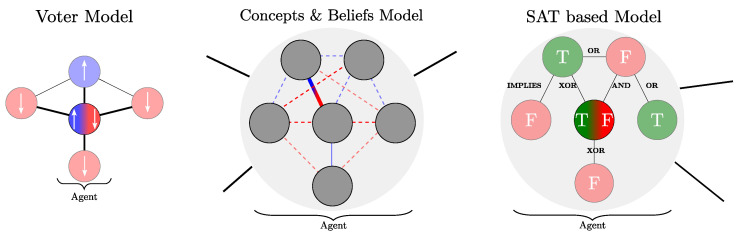
Illustration of three different models of social norm evolution: the voter model (**left**), the concept–belief model (**center**), and our SAT belief model (**right**). In the voter model, an agent is represented by a node, and its opinion (up or down) evolves according to the opinion of the other agents with which it is connected. Here, the agent adopts the opinion of the majority of its neighbors. In the concept–belief model [10], an agent is described by an internal network of concepts (internal nodes) and beliefs (internal links). The links can be positive (blue) or negative (red). The belief network is embedded in a social network, and the agent evolves by modifying the sign of the links to minimize the incoherence of its belief system and the pressure of peer influence. In the SAT model, an agent has a cognitive network of propositions (TRUE or FALSE) connected by logical operators (OR, XOR, AND …). An agent evolves by exchanging, with another agent, the value of a proposition to increase its internal coherency. Importantly, in the SAT model, the topology of the network of interactions is not fixed but constantly changes over time as agents agree more or less on the different propositions.

**Figure 2 entropy-27-00358-f002:**
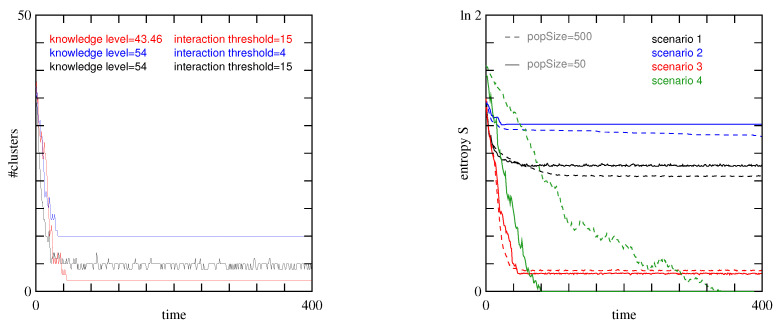
(**Left**): Number of clusters as a function of time for different interaction parameters. In black, any agent will discuss with any other one (scenario 1), whereas in blue, discussion can only take place between agents that do not disagree too much (scenario 2, $r_{int}=4$). The red curve has similar parameters as the black one, except for the level of information of the agents which is not perfect (scenario 3, p=0.8). The figure indicates a *knowledge level* for the red curve. This is the average number of relations in the population, as will be discussed later. (**Right**): Entropy of the propositions values, as a measure of the diversity of the belief systems in the population, for each scenario considered in this study. Agents with random beliefs correspond to a maximal entropy state, whereas a homogeneous population with the same belief system has a null entropy. In this panel, the entropy of populations with N=50 and N=500 agents are compared. We observe a very similar qualitative behavior. For scenario number 4, only the case $r_{int}=15$ is shown.

**Figure 3 entropy-27-00358-f003:**
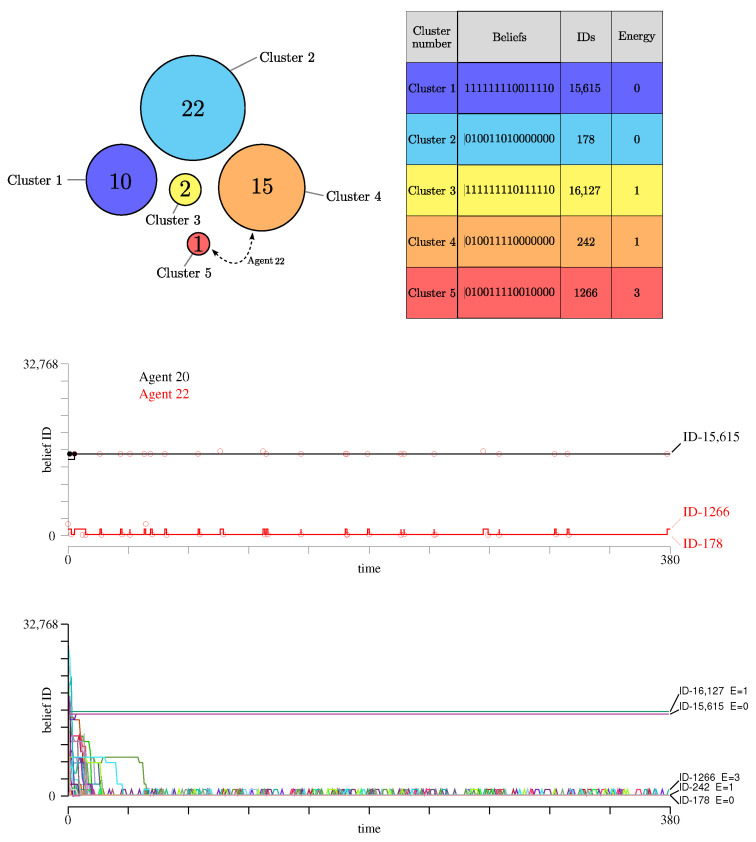
(**Top**): Representation of the clusters of agents emerging in scenario 1. Each cluster is represented by a disk whose diameter is proportional to the number of agents it contains, as indicated by the number in the center. On the right panel, with corresponding colors, the belief system of the agents in each cluster is given with zeros and ones, showing the value of each of the 15 propositions. Belief systems can be interpreted as a binary number whose decimal value is defined as its ID. The last column gives the energy of the belief system. (**Middle**): Trajectory of two selected agents (black for agent 20 and red for agent 22) in the beliefs space. The solid lines show the current belief ID of each agent over time and the circles show the belief ID of the agent responsible for the interaction that caused the change. Note that in this panel, the agent label is not shown, only its belief-ID. (**Bottom**): Trajectory of all agents in the beliefs space, each with a different color chosen at random. The stable clusters are well visible, as well as the agents that do not converge in a final belief system.

**Figure 4 entropy-27-00358-f004:**
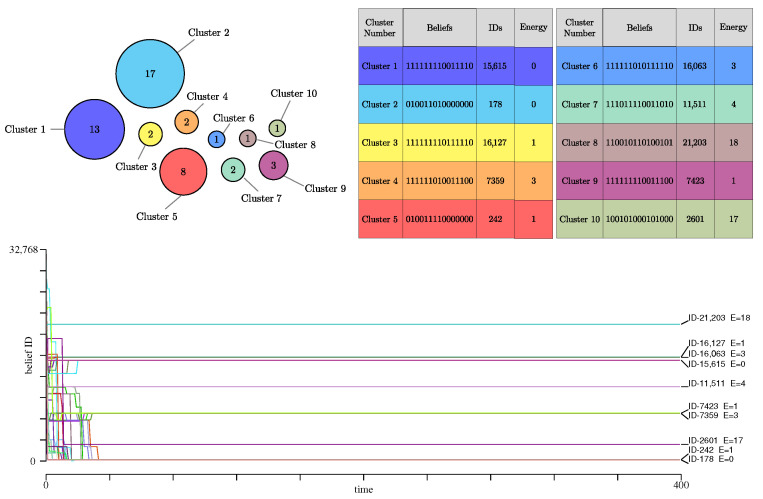
(**Top**): Clusters of agents emerging in scenario 2 ($r_{int}=4$), using the same representation as explained in Figure 3. (**Bottom**): Trajectory of all agents in the beliefs space. As opposed to the case where the interaction range is maximum, here, more clusters are formed, and they are fully stable after a short initial transient regime.

**Figure 5 entropy-27-00358-f005:**
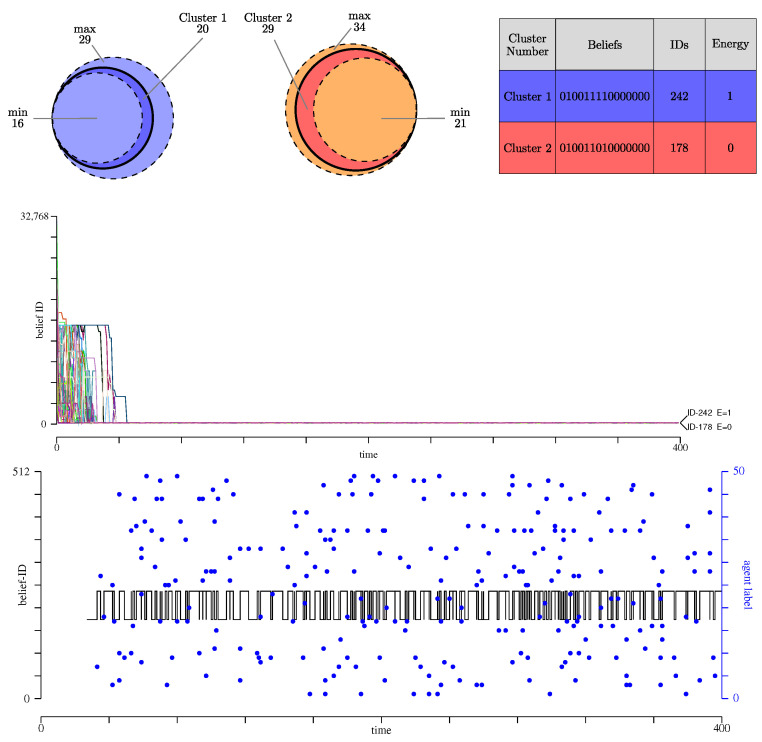
Same representation as defined in Figure 3, now for scenario 3, with partial knowledge (p=0.8) and full interaction range $r_{int}=15$). (**Top**): The two selected clusters and their properties. They are very similar as the only differ by the value of proposition $P_{6}$. When repeating the same experiment with different initial populations, the same clusters are selected but their size may vary from one run to the other, as shown in Table 4. This variation in size is shown here as the fluctuation of the diameter of the disk representing each cluster. (**Middle**): Trajectory of the agent in the beliefs space. (**Bottom**): Trajectory of agent 0 that keeps moving between clusters ID-178 and ID-242 (black curve). The blue circles denote the agent index responsible for the change in cluster, and no longer its cluster ID. The first iterations are not shown here to focus only on the later regime of agent 0.

**Figure 6 entropy-27-00358-f006:**
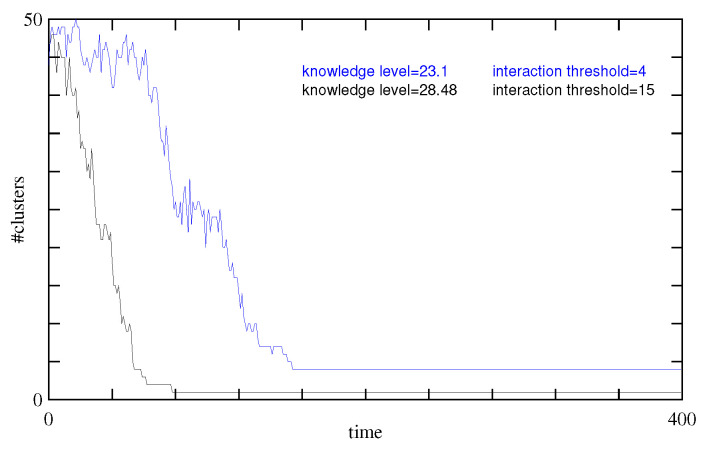
Illustration of the evolution of the number of beliefs clusters in the case of variable knowledge in the population. The knowledge level indicated represents the final average number of relation among the population.

**Figure 7 entropy-27-00358-f007:**
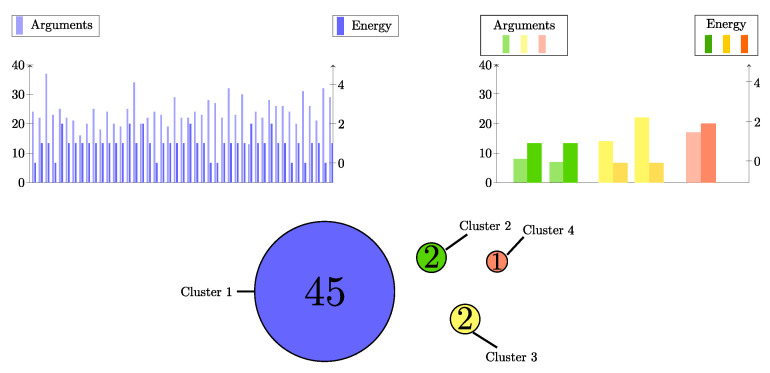
Clusters of beliefs in scenario 4 with $r_{int}=4$. In this scenario, the cognitive matrix $A_{i}$, the energy and knowledge level of each individual may vary. For this reason, the upper panels show for, 45, 2, 2 and 1 agents, in the four clusters, their energy and knowledge level.

**Table 1 entropy-27-00358-t001:** Size (number of individuals) of the clusters that emerge after 400 Monte-Carlo steps in scenario 1, for 6 repetitions of the process, with each time a different initial population, but the same cognitive matrix *G*.

Cluster ID	Run 1	Run 2	Run 3	Run 4	Run 5	Run 6
178	15	16	21	23	18	24
242	27	26	20	21	18	16
1266		2			1	
15615	7	6	9	5	13	9
16127	1					1

**Table 2 entropy-27-00358-t002:** Sizes of the largest clusters observed after 400 Monte-Carlo steps for scenario 1, and two population sizes, N=50 and N=500. The value shown here are the result of one experiment. Within this statistical uncertainty, the sizes of the clusters typically increase by a factor of 10.

Cluster-ID	N=50	N=500
15615	10	111
178	22	163
16127	2	7
242	13	106

**Table 3 entropy-27-00358-t003:** Size of the clusters that emerge after 400 Monte-Carlo steps in scenario 2, for 6 repetitions of the process, with each time a different initial population, but the same global cognitive matrix *G*, and an interaction threshold of $r_{int}=4$. The sum of the columns is not equal to 50, the population size N, because small and non-repetitive clusters are omitted in each run.

Cluster ID	Run 1	Run 2	Run 3	Run 4	Run 5	Run 6
178	11	13	12	20	17	22
242	12	11	6	6	11	9
7359	1	2	3	3	2	
7423	4		5	6	2	5
15551	2	2	3	2	4	
15615	11	13	17	9	10	8
16127	2	1			1	

**Table 4 entropy-27-00358-t004:** Size (number of individuals) of the clusters that emerge after 400 Monte-Carlo steps, for 6 repetitions of the process, with each time a different initial population, but the same fraction p=0.8 of the global cognitive matrix *G* (but different agents may have copied different elements of *G*.)

Cluster ID	Run 1	Run 2	Run 3	Run 4	Run 5	Run 6
178	18	25	17	29	16	19
242	32	25	33	21	34	31

## Data Availability

The C++ code croyances.cc used to produced the results is available on https://cui.unige.ch/~chopard/Beliefs/ (accessed on 26 February 2025). A readme.txt file explains how to run it.

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
