# Peer review of "A Model for the Formation of Beliefs and Social Norms Based on the Satisfaction Problem (SAT)"

_entropy, 2025, doi:10.3390/e27040358_

Round 1
Reviewer 1 Report
Comments and Suggestions for Authors
This paper presents an approach to social interactions which differs from others in the literature and is based on belief systems. The approach is described in detail and some results are presented. Reference is made to other workers who have used different approaches but after presenting their results no comparison with other approaches is provided. This is necessary before publication in my opinion. Other questions I have are: How significant is the individual belief relative to the overall interaction between individuals? How do the results depend on group size? Real life experimental tests have also been made to compare the validity of other methods. Have the authors of this paper conducted similar experiments to assess their approach?
Author Response
See attached pdf file.

Reviewer 2 Report
Comments and Suggestions for Authors
It's amazing how often it happens that scientists proceed in parallel, doing similar things unaware of one another because of disciplinary boundaries. This is one such case.
My review aims at making the authors aware of fellow scientists who did similar things in other disciplines. First of all, the Constraint Satisfaction Networks (CSNs) developed by cognitive scientists. In my opinion, mentioning CSNs somewhere would increase the visibility and relevance of your paper. Moreover, a few additional quotations may better support your choices.
Secondly, a series of psychological experiments showing that humans very early make a choice based on consistency, being attracted by a dominant structure. Thereafter, they spend much time justifying this choice somehow - for instance, by building a straw-man representing a ridiculous version of the opposite option. This may not be directly relevant to this paper, but you may make some usage of it in future research.
Thirdly, I want to make you acquainted with a literature that derives (empirically verified) structural features of the brain from free energy absorption by constructing coherence. Again, not directly relevant, yet somehow related.
Finally, I would like to point to the possibility that collective states characterized by unanimity are unstable. This is my personal opinion and you don't need to agree, and even if you do, it does not concern this paper.
My only strict requirement is that you make your code available in some public repository. You already declared that the code is available upon request, but making it available at a public repository makes its availability independent from the vagaries of life.
You may select whichever repository you prefer, but I would like to call your attention on CoMSES - OpenABM (https://www.comses.net/). It's free and, if you submit your code to a verification procedure, your code is published with a DOI.
OpenABM is meant for agent-based models in social and life sciences so much of the code is written in some object-oriented language, but in my opinion you can safely claim that you have an ABM at a sufficiently high level of abstraction even if you wrote your model in C++. If they object, you may point to the fact that a large fraction of their models have been built on NetLogo, which is not object-oriented either.
Here is the index of my review:
1) Constraint Satisfaction Networks;
2) Experiments on the search for dominance;
3) Cognition as free energy minimization;
4) Polarized minds, polarized worlds;
5) Minor issues.
1) CONSTRAINT SATISFACTION NETWORKS
CSNs are a simpler version of the Rodriguez, Bollen and Ahn (2016) paper that you cite. While Bollen et al. represent individuals as networks of concepts, and then society as a network of networks, CSNs are simple networks of concepts that either excite or inhibit one another. In my understanding, your model can be seen as an extension of Rodriguez et al.'s where implication is not the only logical relation.
CSNs were mainly invented in the 1980s and 90s. They typically generated polarization of opinions. Here is a selection of publications that sum up their main results:
- Thagard, P. (1989) Explanatory Coherence. Behavioral and Brain Sciences, 12: 435-502.
- Thagard, P. (2000) Coherence in Thought and Action. The MIT Press.
Here are a few more recent papers reporting on experiments where decision-makers were probed at intermediate stages of the decision process. The decision process was eventually reconstructed by means of CSNs:
- Holyoak, K.J. and Simon, D. (1999) Bidirectional Reasoning in Decision Making by Constraint Satisfaction. Journal of Experimental Psychology: General, 128: 3-31.
- Simon, D. and Holyoak, K.J. (2002) Structural Dynamics of Cognition: From consistency theories to constraint satisfaction. Personality and Social Psychology Review, 6: 283-294.
- Lundberg, C.G. (2007) Models of Emerging Contexts in Risky and Complex Decision Settings. European Journal of Operational Research, 177: 1363-1374.
- Gloeckner, A., Betsch, T. and Schindler, N. (2010) Coherence Shifts in Probabilistic Inference Tasks. Journal of Behavioral Decision Making, 23: 439-462.
Here are a few papers that somehow attempt to go beyond simple CSNs in order to model interacting individuals. Some of them are purely conceptual, others are empirical, still others attempt to design a network. In various respects, they are related to your and Rodriguez et al.'s papers:
- Schulz-Hardt, S., Frey, D., Luethgens, C. and Moscovici, S. (2000) Biased Information Search in Group Decision Making. Journal of Personality and Social Psychology, 78: 655-669.
- Polman, E. (2010) Information Distortion in Self-Other Decision Making. Journal of Experimental Social Psychology, 46: 432-435.
- Bhatia, S. and Golman, R. (2019) Bidirectional Constraint Satisfaction in Rational Strategic Decision Making. Journal of Mathematical Psychology, 88: 48-57.
- Dalege, J., Galesic, M. and Olsson, H. (2024) Networks of Beliefs: An integrative theory of individual and social-level belief dynamics. Psychological Review, DOI 10.1037/rev0000494.
- Smart, P.R. and Shadbolt, N.R. Modelling the Dynamics of Team Sensemaking: A constraint satisfaction approach. Univ. of Southhampton, School of Electronics and Computer Science, working paper.
There are very many publications linking CSNs to Gestalt Theory or the philosophy of science. You may rreach them by looking for the many publications of Paul Thagard. Here below I list a few books or papers that in my opinion are interesting because (1) They offer detailed empirical examples, and (2) They often imply networks that add or create nodes. Thus, they offer hints for better modelizations:
- Thagard, P. (1999) How Scientists Explain Disease. Princeton University Press.
- Gloeckner, A. and Betsch, T. (2008) Modeling Option and Strategy Choices with Connectionist Networks: Towards an integrative model of automatic and deliberate decision making.
- Mosier, K.L. (2009) Searching for Coherence in a Correspondence World. Judgment and Decision Making, 4: 154-163.
2) EXPERIMENTS ON THE ATTRACTION OF THE DOMINANT STRUCTURE
In the 1950s, Festinger introduced in psychology the theory of cognitive dissonance (TCD). TCD claims that, *after* decision has been made, humans justify it by distorting the evidence and the reasons for their choice. However, CSNs suggest that humans construct coherence *before* decision. This is clearly inconsistent with Festinger. Here are a few papers on this:
- Simon, D., Pham, L.B., Quang, A.L. and Holyoak, K.J. (2001) The Emergence of Coherence Over the Course of Decision Making. Journal of Experimental Psychology: Learning, Memory, and Cognition, 27: 1250-1260.
- Simon, D., Snow, C.J. and Read, S.J. (2004) The Redux of Cognitive Consistency Theories: Evidence judgments by constraint satisfaction. Journal of Personality and Social Psychology, 86: 814-837.
- Simon, D., Krawczyk, D.C. and Holyoak, K.J. (2004) Construction of Preferences by Constraint Satisfaction. Psychological Science, 15: 331-336.
It's a pity that researchers working on CSNs were apparently unaware of a series of experiments carried out in the 1980s by Henry Montgomery and Tadeusz Tyszka where they were able to show that decision takes place very early, as soon as the brain is attracted by the coherence of one possible explanation. The rest is justification. It is clear that Montgomery and Tyszka's experiments were opposed by most other psychologists, because they mainly published their papers as book chapters. However, their findings are still important because they show that sometimes, in order to justify the interpretation that we find most attractive, we build a straw-man of the opposite interpretation, making it ridicule to the point that we can easily dismiss it. Here is a selection of their publications:
- Montgomery, H. (1983) Decision Rules and the Search for a Dominance Structure: Towards a process model of decision making. In P. Humphreys, O. Svenson, A. Vary (eds.) Advances in Psychology, Vol. 14: Abalysing and Aiding Decision Processes, pp. 343-369. North-Holland.
- Tyszka, T. (1983) Contextual Multiattribute Decision Rules. In L. Sjoberg, T. Tyszka, J.A. Wise (eds.) Human Decision Making, pp. 243-256. Bodafors, Doxa.
- Montgomery, H. (1989) From Cognition to Action: The search for dominance in decision making. In H. Montgomery and O. Svenson (eds.) Human Decision Making, pp. 23-49. John Wiley & Sons.
- Montgomery, H. and Svenson, O. (1989) A Think-Aloud Study of Dominance Structuring in Decision Processes. In H. Montgomery and O. Svenson (eds.) Human Decision Making, pp. 135-150. John Wiley & Sons.
- Montgomery, H. and Willen, H. (1999) Decision Making and Action: The search for a good structure. In P. Juslin and H. Montgomery (eds.) Judgment and Decision Making: Neo_Brunswikian and Process-Tracing Approaches, pp. 147-173. Lawrence Erlbaum Associates.
4) COGNITION AS FREE ENERGY MINIMIZATION
Karl Friston has a long series of papers where he understands cognition as free energy minimization, with living being selecting those stimuli that increase coherence with their mental representations. Friston has been able to explain certain stuctural features of the human brain. Recently, his insights have merged with humanists'. Here is a tiny small selection out of an immense literature:
- Friston, K., Kilner, J. and Harrison, L. (2006) A Free Energy Principle for the Brain. J. of Physiology Paris, 100: 70-87.
- Friston, K. (2008) Hierarchical Models in the Brain. PLOS Computational Biology, 4 (11): e1000211.
- Friston, K.J. and Frith, C. (2015) Active Inference, Communication and Hermeneutics. Cortex, 68: 129-143.
- Palacios, E.R., Razi, A., Parr, T., Kirchhoff, M. and Friston, K. (2020) On Markov Blankets and Hierarchical Self-Organization. J. of Theoretical Biology, 486: 110089.
- Veissiere, S.P.L., Constant, A., Ramstead, M.J.D., Friston, K. and Kirmeayer, L.J. (2020) Thinking Through Other Minds: A variational approach to cognition and culture. Behavioral and Brain Sciences, 43, e90: 1-75.
4) POLARIZED MINDS, POLARIZED WORLDS
You claim that, under certain conditions, one single opinion spreads. While I agree that this dynamics exists, I also think that as soon as one single opinion has conquered the world, endogenous processes - perhaps triggered by random fluctuations and local coherence - make one alternative opinion emerge. You don't need to agree with me, but I rather have the impression that bi-polarism, or perhaps bi-polarism + smaller opinion islands, is the deepest attractor. Therefore, it exhibits greater meta-stability.
In my opinion, this happens at individual as well at the social level. In any village you will find two factions. In regional and national politics, you find two main parties, or two coalitions of parties, perhaps with a few smaller parties moving the equilibrium towards one or the other side. Even when a dictator succeeds to monopolize the scene, absolute dominance doesn't last for long. Hitler had reached absolute control, but he stayed in power just 11 years; Mussolini stayed in power long enough to see fascism (unofficially) splitting into several streams (e.g., those who wanted to ally with Germany vs. those who wanted to ally with the Great Britain); Stalin killed Trotzky, but Krushov had been silently building up a counter-power, and so on and so on. The Roman Empire had managed to conquer (almost) the whole known world, so they split it in two. Jump a couple of millennia, the U.S.A. reached world supremacy in 1989, to find out within a couple of decades that China and Russia had emerged as global competitors.
This is not a request for changing your paper. Just an exchange of opinions. In your paper, you do find that the two-opinions state appears very often. Then you write about the danger that one single opinion monopolises the scene. In my opinion, the opposite dynamics exists as well. So you see, even you and me, we developed two opinions!
5) MINOR ISSUES
You assumed that individuals ignore opinions that are too different from theirs. This is known as "cognitive distance." Interaction is meaningful when cognitive distance is intermediate. If it's too low, there's nothing to learn. If it's too high, learning is impossible because one doesn't understand what the other says. This is, I think, the basic reference:
- Nooteboom, B. (2000) Learning and Innovation in Organizations and Economies. OUP.
You mention the "metropolis" thing as if everyone would know it, but I did not. Later on, you write a formula that enables readers to understand what metropolis is. Yes, I knew it, but not with this name.
Author Response
See attached pdf file

Round 2
Reviewer 1 Report
Comments and Suggestions for Authors
I am content to recommend publication